# Fly, Fail, Fix: Iterative Game Repair with Reinforcement Learning and Large Multimodal Models

**Alex Zook, Josef Spjut, Jonathan Tremblay**

{azook,jspjut,jtremblay}@nvidia.com

**NVIDIA**

## Abstract

Game design hinges on understanding how static rules and content translate into dynamic player behavior—something modern generative systems that inspect only a game's code or assets struggle to capture. We present an automated design iteration framework that closes this gap by pairing a reinforcement learning (RL) agent, which playtests the game, with a large multimodal model (LMM), which revises the game based on what the agent does. In each loop the RL player completes several episodes, producing (i) numerical play metrics and/or (ii) a compact image strip summarising recent video frames. The LMM designer receives a gameplay goal and the current game configuration, analyses the play traces, and edits the configuration to steer future behaviour toward the goal. We demonstrate results that LMMs can reason over behavioral traces supplied by RL agents to iteratively refine game mechanics, pointing toward practical, scalable tools for AI-assisted game design.

## 1 Introduction

Game design is an iterative process: designers create a game and have players playtest the game to provide feedback for further refinement of the game design (Figure 1). Playtesting helps designers understand how statically authored rules and content produce dynamic gameplay behavior when players interact with that static content. The complex, emergent dynamics that result from players engaging with a game makes it difficult to reason about a design solely from the rules and content.

Here we explore the use of large multimodal models (LMMs) for the task of iteratively refining a game design using player behavior. LMMs are increasingly used to generate games (Todd et al., 2023; Sudhakaran et al., 2023; Anjum et al., 2024; Zala* et al., 2024), yet ensuring the game yields desired player behaviors remains difficult (Sun et al., 2024), in part due to the difficulty of reasoning about games purely from their static description as rules and content.[1] Reinforcement learning (RL) agents have demonstrated strong game playing capabilities across many genres (Mnih et al., 2015; Silver et al., 2018; Hafner et al., 2025; Vinyals et al., 2019; Berner et al., 2019). While having humans playtest a game is the most direct way to gather human feedback (Zook et al., 2014), this can be costly and time-consuming to implement. We investigate an alternative where an RL agent acts as a proxy for a human player. In our iterative design process, an LMM takes the role of the designer modifying the game, using gameplay behavior from the RL player to guide decisions. We use this setup to explore the potential for AI to augment the design iteration process by automatically refining a game design toward a given gameplay goal.

We test this approach in Flappy Bird, fixing broken level generators to achieve a target player score and using a pretrained DQN agent as the player. We explore two different representations of player behavior to the LMM designer: textual summaries of gameplay metrics captured from the game and

---

[1]Please consult Appendix A for a more complete description of related work.

visual summaries of gameplay extracted from video recordings. Our experiments demonstrate the viability of automated design iteration using LMMs, showing equal levels of LMM success at tuning gameplay difficulty when using text-based metrics, gameplay behavior visuals, or both together. These results showcase the value of RL agents to facilitate automated design iteration, enabling the refinement of games to provide desired interactive experiences through automated modification by LMM agents.

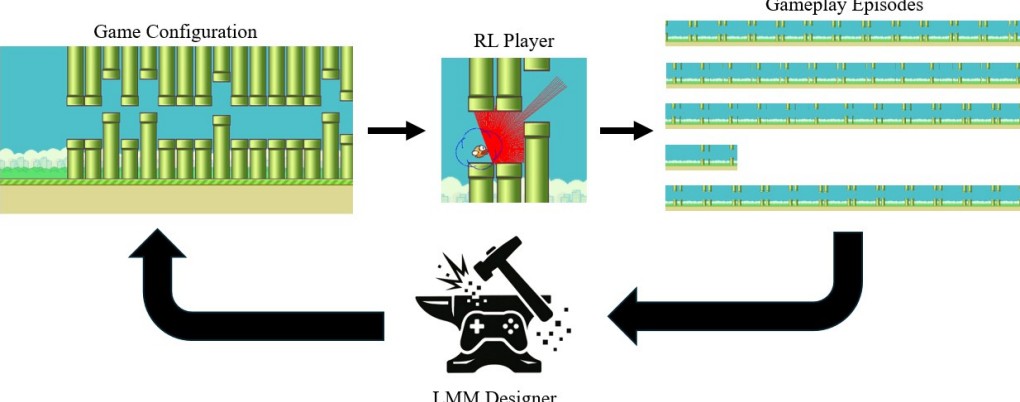

Figure 1: Overview of our AI design iteration framework. In each iteration an RL agent plays a given game configuration, yielding gameplay behavior traces that an LMM uses to modify the game configuration. We vary the play trace representation as summary metrics, or a sequence of image observations, or both.

## 2 Method

Our iterative design system consists of designer and player agents. The *player* is an RL agent that plays the game to produce information about gameplay behavior possible in the game. The *designer* is an LMM that is provided a gameplay goal and reasons about the current game configuration and the gameplay behavior to modify the game to achieve the gameplay goal. Through an iterative process, the designer uses the information extracted from play sessions to make changes about the game, and the player then plays the new game design to provide feedback for a new iteration.

**Game**    We study Flappy Bird for it's accessibility and availability of trained RL agents. Flappy Bird's gameplay is governed by the physics of the game (rules) and placement logic for the procedurally generated pipes (content), making it difficult to reason about purely from a textual description of the game rules and thus a good testbed for our iterative design process. Levels are procedurally generated, randomizing the setup for each playthrough. We instrumented the game to record key gameplay metrics including the player score (number of pipes passed) and gameplay duration.[2] Each playthrough was also recorded as a video clip.

**Agents**    For the *designer* we used GPT-4.1 (https://openai.com/index/gpt-4-1/) as it is currently a strong LMM model that demonstrates strong baseline visual reasoning capabilities. The designer task is to modify a configuration file that defines parameters of key game mechanics including pipe spacing, dimensions, movement speed (see Appendix B for detailed information).[3] Concretely, in each iteration the designer is prompted with the gameplay goal having a score of 10 (no more nor

---

[2]While we recorded other metrics, we observed the LMM would only reference these features in it's analysis, suggesting it was choosing to ignore the less relevant metrics like the maximum height in the screen the player flew to.

[3]We briefly investigated local LMMs, but found their performance to currently be relatively weak at basic tasks including producing valid configuration file outputs. Given the rapid pace of improvements in LMMs we anticipate these limitations will be relieved in the relatively near future and plan to revisit open models at that time, see Appendix D.

less), the current game configuration (as YAML), a text description of the configuration parameter semantics, and examples of several (5) play trajectories from the game.

For the *player* we used Dirgová Luptáková et al. (2024) DQN implementation as it offers an agent that is stable to game variations.[4] This RL agent uses a lidar model to observe the environment and decides when to make the bird flap (jump); we found this model is flexible to variation to the level design. In our experiments the players produces 5 episodes of play from each configuration from the designer. For our experiments we use the pretrained agent without modification, leaving agent tuning and learning to future work.

## 3   Experiments

We evaluate four experimental conditions to assess the impact of different feedback modalities provided to the LMM for game configuration adjustment:

1. **Config-only:** Designer receives only the configuration file and its description.
2. **Text-only:** Adds summaries for 5 episodes (score and flight time).
3. **Image-only:** Adds one composite image per episode from the last 8s of gameplay (25 frames).
4. **Text+Image (Ours):** Combines textual summaries and composite images.

In all experimental conditions, we prompt the designer to adjust the game configuration with the objective of achieving a target score of 10, corresponding to the agent successfully passing 10 pipes. An episode is considered complete when the agent collides with a pipe, reaches a maximum duration of 120 seconds, or attains a maximum score of 30, whichever occurs first. To test reasoning across a variety of game design conditions we created five starting game configurations for the LMM to adjust (see appendix B).

A single trial run consists of the initial broken configurations, followed by 9 sequential iterations of changes to the configuration for a total of 10 configurations, see Figure 3 and Appendix E. For each configuration, we collect 5 independent player episodes. The LMM then uses text metrics and/or the images from these 5 episodes to revise the game configuration. This process of episode collection and configuration adjustment is repeated for 9 iterations in a row, each resuming from the previous configuration, allowing the model to iteratively refine the game settings. We conduct 10 independent trials for each experimental condition for each game configuration.

We analyze the DQN following the recommendations from the RL statistical best practices literature (Agarwal et al., 2021), using the inter-quartile mean (IQM) with 5000 bootstrap samples to estimate 95% confidence intervals. Sampling 5 episodes from each configuration leads to high variance in estimates of the gameplay behavior. Thus, for analysis, we ran 50 additional episodes for every configuration that was generated during the iterations. These trials use the same configuration as the original trial, thereby retaining identical game dynamics while providing extra data for evaluation purposes; we only analyze this newly generated data. Figure 2 displays the DQN score (IQM and 95% CI) across different iterations of configuration changes.

The configuration-only baseline fails to improve the game settings, resulting in the agent consistently achieving a score of zero. Text-based feedback reliably allows the designer to achieve the target objective, with scores hovering around 10. Image-based feedback serves equally well, highlighting capability of current LMMs to reason about visual representations of gameplay, at least in our case where score is easily discerned from progress in the level. Providing both text and image feedback also allows accomplishing the task by the 10th iteration. We note that all 3 non-baseline models have statistically indistinguishable performance by the 10th iteration, and often achieve the target score in fewer iterations (often the 5th). These findings collectively indicate that LMMs can reason about gameplay behavior and relate this to game design parameters to achieve specified gameplay objectives.

---

[4] github.com/markub3327/flappy-bird-gymnasium

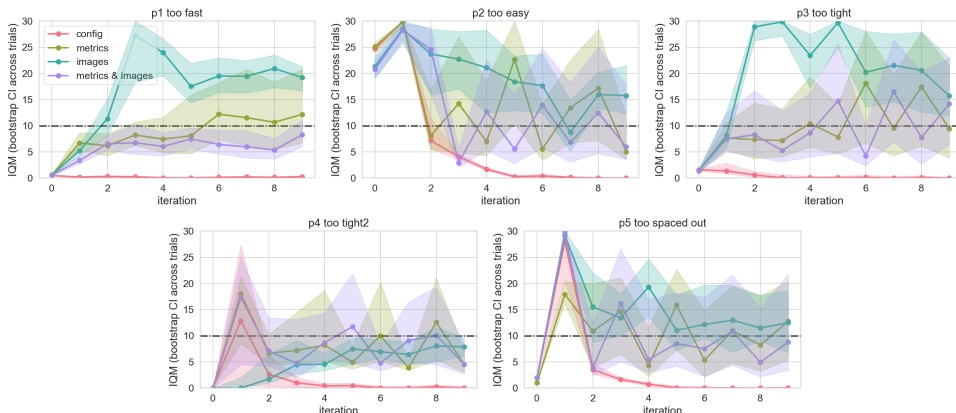

Figure 2: Player score (IQM and 95% CI) across iterations from different starting configurations. The dotted line is the target score for the designer.

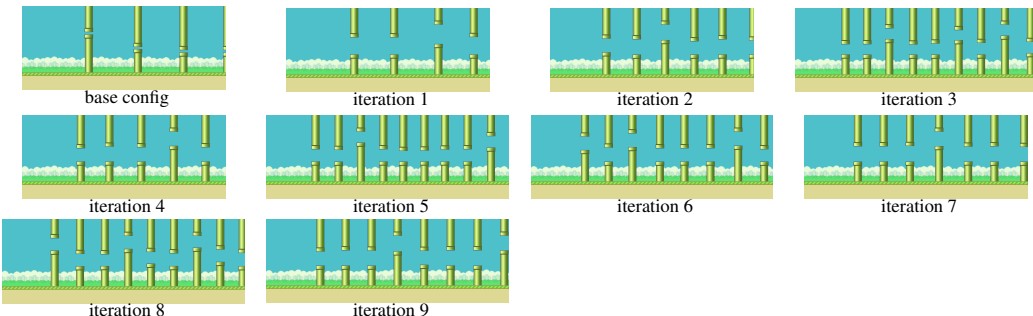

Figure 3: Designer steps from the base config (top left) to the final generated config (bottom right). Each level represents ~8 seconds of play. Levels are procedurally generated in Flappy Bird, thus different game speeds generate different length levels.

## 4    Conclusion

This study showcases the value of RL agents as playtesters in an automated design iteration loop with an LMM responsible for adjusting game parameters to achieve a gameplay objective using the RL agent's behavior as feedback. We showed that providing text or image feedback to the LMM designer allows it to achieve the target score after at most 10 iterations of refinement. These results open the way for further investigation of automated game iteration and refinement using a variety of play behavior modalities.

**Future Work.**   Our study opens several avenues for further investigation. First, in preliminary experiments we found that minor change to the player physics parameters (acceleration applied by jumping, gravity strength, and so on) frequently cause catastrophic degradation in agent performance despite the resulting games remaining playable to humans. Addressing this brittleness is critical for building automated design loops that simultaneously produce robust RL agents and diverse training environments. Second, we see strong potential in replacing the single, fixed RL player with an ensemble of agents with heterogeneous architectures (both learning and static), providing a richer proxy for the diversity of human players. Third, we plan to enlarge the designer's action space from configuration file edits to modification of the game code, enabling the generation of new mechanics and gameplay.

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

# Supplementary Materials

*The following content was not necessarily subject to peer review.*

## A   Related Work

**LLM-Generated Simulations.**   LLMs have been widely used to generate various simulation components, including environment configurations, reward functions, and tasks for both games and robotics. In games, LLMs have been used to generate levels (Todd et al., 2023; Sudhakaran et al., 2023), parameterize simulators (Zala* et al., 2024), and assist in full-game design (Anjum et al., 2024). Recently, FactorSim demonstrated that LLMs are capable of generating full fledge video games from a detailed game design document while passing specific unit tests (Sun et al., 2024), however they did not assure that the generated games are playable. In robotics, systems such as RoboGen (Wang et al., 2024), Holodeck (Yang et al., 2024), and Gen2Sim (Katara et al., 2024) have used LLMs to create 3D environments, tasks, and interactive scenes, often leveraging multi-prompt pipelines. Other efforts have focused on reward design (Ma et al., 2023; Kwon et al., 2023; Ma et al., 2024), with Eureka (Ma et al., 2023) using agent performance feedback to iteratively refine task rewards.

By contrast, we use the raw behavior trajectories of an RL agent as input to a generic LMM, either as telemetry summary data or raw video. We demonstrate that visual data from gameplay alone is sufficient for the agent to refine game parameters, without additional processing of that format. This allows the AI game designer to iteratively refine simulation structure based on observed gameplay, reducing the need for human engineering to guide the system and opening the door to integrating this approach into game development workflows where telemetry is difficult or impractical to implement.

We believe our work provides a foundation for interpretable game design iteration by LMMs, which can later be extended to collaborate with human designers. Unlike POET (Wang et al., 2019), which does not explain its environment modifications, our LMM-driven approach has the potential to surface rationales for design choices—enabling human-in-the-loop refinement in future systems.

**Reinforcement Learning in Video Games.**   Reinforcement learning (RL) has been extensively applied to video games, serving both as a benchmark for evaluating general intelligence and as a practical tool for developing adaptive agents. Mnih et al. (2015) demonstrated that Deep Q-Networks (DQN) could learn to play a variety of Atari games directly from pixel inputs, achieving human-level performance in several instances. Subsequent work demonstrated that RL agents could be trained to play a variety of games, including complex titles like Dota 2 (Berner et al., 2019) and StarCraft (Vinyals et al., 2019). We complement these efforts by showing how to use RL agent behavior as feedback to iteratively refine game design, providing a new avenue for integrating RL into game development workflows.

A parallel line of work has investigated the joint training of RL agents and construction of learning (game) environments, broadly known as open-ended learning or automated curricula. Stooke et al. (2021) introduced an open-ended learning framework that trains agents across a procedurally generated universe of tasks, encompassing both cooperative and competitive games. Gisslén et al. (2021) take a similar approach, using RL for the an inner loop player and outer loop designer in 3d platform traversal and road driving tasks. Khalifa et al. (2020) addresses 2d puzzle level layout using RL training for the puzzle generation and using fixed solvers (thus omitting the automated curriculum aspect). The POET system and related efforts demonstrated the joint training of an RL agent and the progressive complexification of the environment (Wang et al., 2019; 2020; Samvelyan et al., 2023).

Unlike these efforts, we directly integrate with an existing game and focus on gameplay goals, rather than agent learning progress. We demonstrate that LMMs already possess design iteration capabilities and can be very sample efficient in refining game designs to achieve a gameplay goal, requiring fewer than 10 iterations, each needing only 5 examples of gameplay behavior.

## B Config Example

```
# Speed and Acceleration
speed:
  pipe_vel_x: -4
  player:
    max_vel_y: 10  # max vel along Y, max descend speed
    min_vel_y: -8  # min vel along Y, max ascend speed
    acc_y: 1       # players downward acceleration
    vel_rot: 3     # angular speed
    flap_acc: -9   # players speed on flapping
    rot_thr: 20    # Player's rotation threshold

# Dimensions
dimensions:
  player:
    width: 34
    height: 24
    private_zone: 100  # Radius of the private zone for LIDAR. DO NOT MODIFY.

  lidar:
    max_distance: 200  # Maximum distance for LIDAR rays. DO NOT MODIFY.

  pipe:
    width: 52
    height: 320
    min_gap: 100
    max_gap: 150
    min_gap_distance: 50  # Minimum distance from ground to pipe gap
    max_gap_distance: 150  # Maximum distance from ground to pipe gap
    min_horizontal_spacing: 200  # Minimum horizontal spacing between pipes
    max_horizontal_spacing: 300  # Maximum horizontal spacing between pipes

  base:
    width: 336
    height: 112

  background:
    width: 288
    height: 512
    fill_color: [200, 200, 200]  # RGB color tuple

metrics:
  save_path: "metrics"
  save_on_reset: True
```

We used 5 starting configurations in our experiments. Each was created by modifying the default game parameters to be broken in different ways that the designer would need to fix:

1. *Too fast*: Pipes moved too fast, making the game too hard.

2. *Too easy*: Pipe gaps were too wide, making the game too easy.

3. *Too tight 1*: Pipe gaps were similar to the default, but made too narrow, making the game too hard.

4. *Too tight 2*: Pipe gaps were distributed differently to the default and also too narrow, making the game too hard.

5. *Too spaced out*: Pipes were placed infrequently, causing wide horizontal gaps, making the game too easy.

## C Prompts

```
common_prefix = (
    "You are a game designer tasked with improving the difficulty of a Flappy Bird
        game. "
```

```
    "Your goal is to modify game configuration so the game is challenging but not
        excessively difficult.\n\n"
)

common_suffix = (
    "SECOND, provide the *complete* YAML for the new configuration, enclosed in a
        markdown fenced code block like:\n"
    "```yaml\n<your yaml here>\n```\n"
    "The goal is to arrive at a good configuration with as few attempts as
        possible.\n"
    "Do not modify the LIDAR parameters."
    "Do not modify the player speed parameters. Only modify the parameters related
        to the pipes, including 'pipe_vel_x'."
)

if input_variant == "config_only":
    intro = (
        common_prefix +
        "Below you will find (1) a *schema* describing every configuration
            parameter, and (2) the *current* configuration.\n\n"
        "First, ANALYSE the configuration and explain (succinctly) what changes
            would improve gameplay.\n" +
        common_suffix
    )
elif input_variant == "images_only":
    intro = (
        common_prefix +
        "Below you will find (1) a *schema* describing every configuration
            parameter, (2) the *current* configuration, and (3) a set of gameplay
            snapshots from recent sessions.\n\n"
        # "Sessions are to passing 30 pipes, while passing fewer than 4 is
            considered too difficult.\n\n"
        "Aim for passing 10 pipes."
        "First, ANALYSE the configuration and images and explain (succinctly) the
            current level of difficulty and what changes would improve gameplay.\n
            " +
        common_suffix
    )
elif input_variant == "metrics_text":
    intro = (
        common_prefix +
        "Below you will find (1) a *schema* describing every configuration
            parameter, (2) the *current* configuration, and (3) a handful of
            recent game-session metrics.\n\n"
        # "Sessions are limited to a maximum score of 30, while a score below 4 is
            considered too difficult.\n\n"
        "Aim for a score of 10."
        "First, ANALYSE the configuration and metrics (paying special attention to
            the recorded scores) and explain (succinctly) the current level of
            difficulty and what changes would improve gameplay.\n" +
        common_suffix
    )
else:  # metrics_and_images
    intro = (
        common_prefix +
        "Below you will find (1) a *schema* describing every configuration
            parameter, (2) the *current* configuration, and (3) recent game-
            session metrics together with gameplay snapshots.\n\n"
        # "Sessions are limited to a maximum score of 30, while a score below 4 is
            considered too difficult.\n\n"
        "Aim for a score of 10."
        "First, ANALYSE the configuration and metrics and explain (succinctly) the
            current level of difficulty and what changes would improve gameplay.\
            n" +
        common_suffix
    )

if input_variant == "config_only":
```

```
    user_content_header = (
        "Configuration schema (read-only):\n" + schema_description + "\n\n" +
        "Base configuration (YAML):\n" + base_yaml
    )
elif input_variant == "metrics_text":
    user_content_header = (
        "Configuration schema (read-only):\n" + schema_description + "\n\n" +
        "Base configuration (YAML):\n" + base_yaml + "\n\n" +
        f"Below you will find up to {n_recent} recent session metrics."
    )
elif input_variant == "images_only":
    user_content_header = (
        "Configuration schema (read-only):\n" + schema_description + "\n\n" +
        "Base configuration (YAML):\n" + base_yaml + "\n\n" +
        f"Below you will find up to {n_recent} gameplay snapshots from recent
            sessions."
    )
else:  # metrics_and_images
    user_content_header = (
        "Configuration schema (read-only):\n" + schema_description + "\n\n" +
        "Base configuration (YAML):\n" + base_yaml + "\n\n" +
        f"Below you will find up to {n_recent} recent session metrics, each
            followed by a gameplay snapshot."
    )

messages = [
    {"role": "system", "content": intro},
    {"role": "user", "content": user_content_header},
]
```

## D   Open Models

While open models did not perform as well as the OpenAI models we did test a variety of text-only and text + image models. We used ollama as our run time and tested many variations of qwen3, deepseek-r1, llama3.2, gemma3, mistral-small3.1, and phi4. Some models inexplicably struggled to consistently generate valid yaml configurations, so we excluded them (all qwen3 variants, deepseek-r1, and llama3.2). We noticed the multi-modal models stopped generating valid yaml when the same size images were used to prompt as in the closed model testing, so we also excluded any image-based testing. As a result, we only generated results on the metrics-only case for a set of 3 open models, gemma3, mistral-small3.1, and phi4.

We share the aggregate results across the same number of iterations as the closed model in Figure 4. We only ran 5 independent trials per model and starting configuration in contrast to the 10 independent trials from Figure 2. Note that the results were consistently lower quality than the closed model, and tended to undershoot the target score, meaning that the game remained too difficult in these cases. Expanding on these open model investigations is left to future work, though we note that the rate of model improvements suggests that near-future models will be significantly more capable.

## E   Level iteration examples

Below are examples of the generated levels over iteration steps (baseline at the top, subsequent iterations are each row below) for each of the configurations. As levels are procedurally generated in Flappy Bird, we display levels corresponding to the length of time the player agent progressed through the level before failure (or termination from level length or timeout).

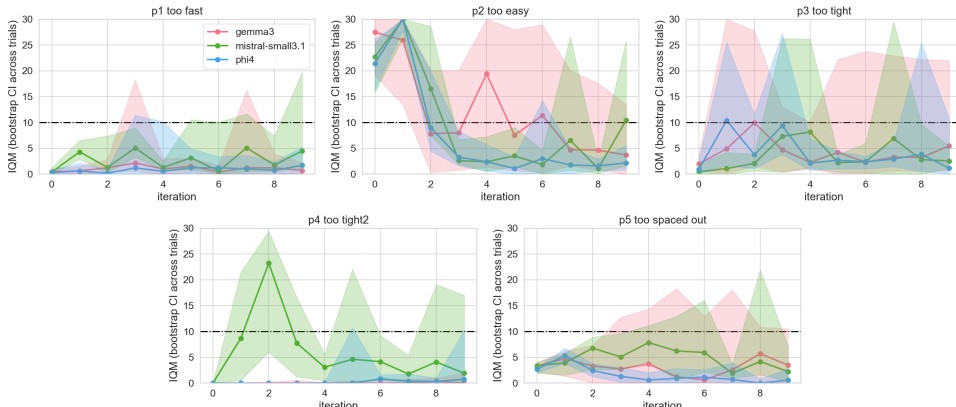

Figure 4: Player score (IQM and 95% CI) across different iterations of configuration changes. Iteration 0 is the initial broken configuration.

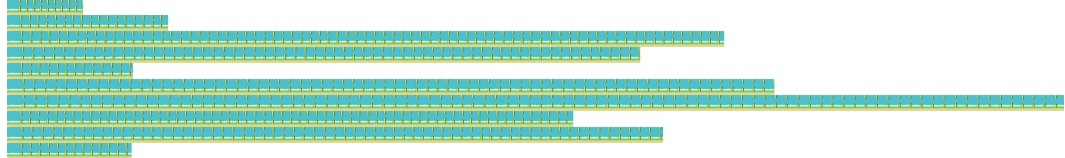

Figure 5: Too fast configuration

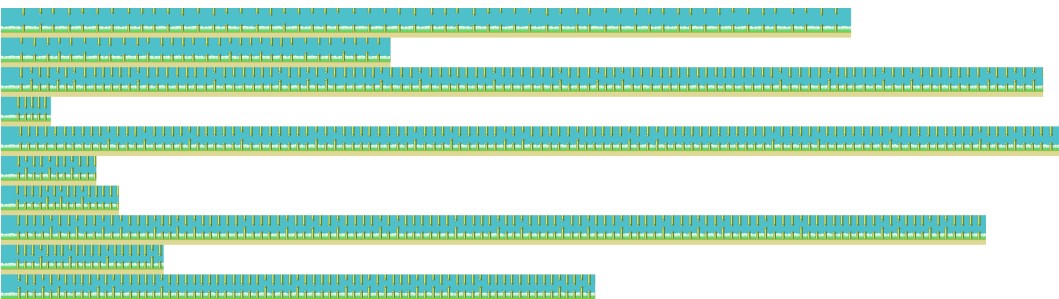

Figure 6: Too easy configuration

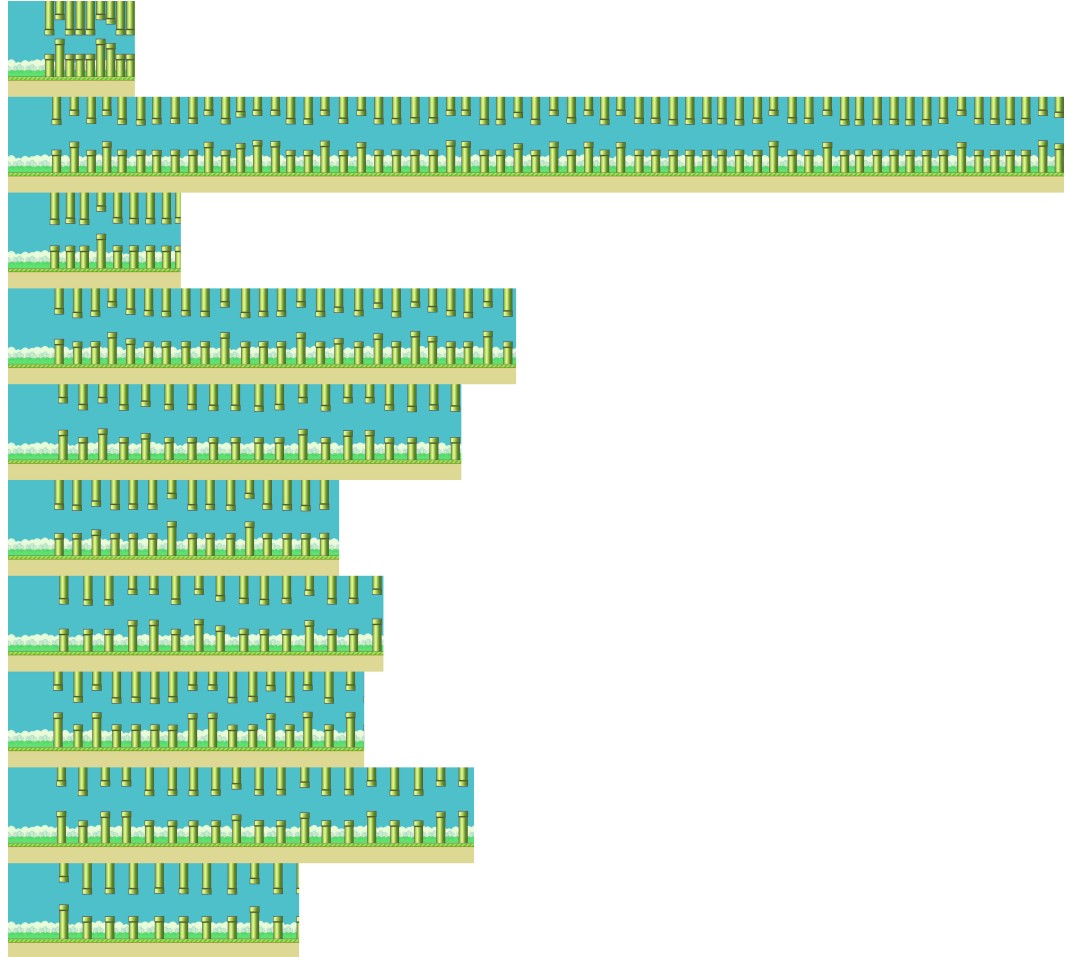

Figure 7: Too tight 1 configuration

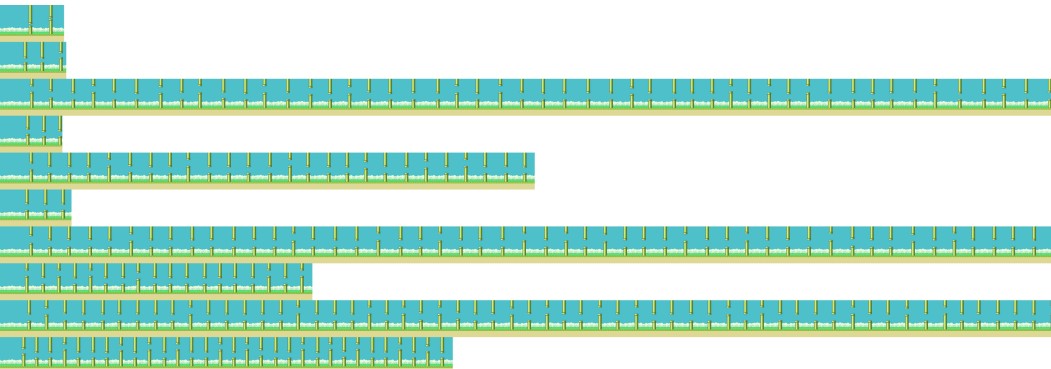

Figure 8: Too tight 2 configuration

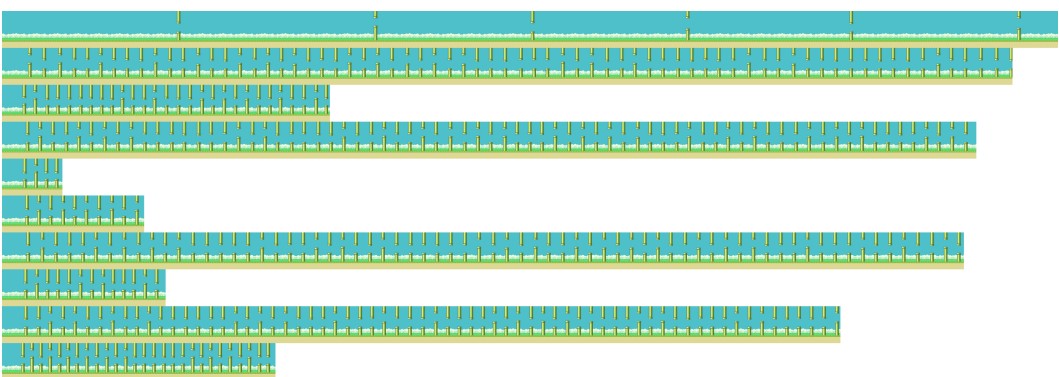

Figure 9: Too spaced out configuration