# OpenReview forum: "Fly, Fail, Fix: Iterative Game Repair with Reinforcement Learning and Large Multimodal Models"
_rl-conference.cc/RLC/2025/Workshop/RLVG — RLVG Workshop - RLC 2025_

### Official Review · Reviewer_2dMK · 2025-06-15
**Interesting approach but the claims are not well supported**

**Rating:** 2
**Confidence:** 4

**Summary:**

The paper proposes a new framework for automated design iteration, where a reinforcement learning (RL) agent is trained to solve a level while a large multimodal model (LMM) revises the game based on the performance of the RL agent. The LMM receives a gameplay goal and modifies the level accordingly. The authors test different modalities (text, text without metrics, images, text + images) and validate the performance of the approach in a Flappy Bird environment.

**Strengths:**

The approach is interesting and I think a good fit for the workshop. The experiments that the authors do, although not complete in my opinion, are interesting and prove that the approach has potential. Moreover, the approach could be really used by real game designers as an assistant automated designer framework. The paper is generally well written and easy to follow.

**Weaknesses:**

*What are the main weaknesses of the paper? Please also state how the authors could address these weaknesses.*

Other than the weaknesses described in the *Claim* section, I think the paper lacks some important details:

1. There is a growing body of literature that studies how RL agents can be used in tandem with other agents to improve level design, but this literature is not even mentioned in the Related Work section. Some examples: Pcgrl: Procedural content generation via reinforcement learning, Khalifa et al., 2020; Adversarial reinforcement learning for procedural content generation, Gisslén et al., 2021.
2. There are many important descriptions that are not present in the paper, but I believe they are important to fully understand (and replicate) the paper. In particular:
    a. The description of the game Flappy Bird. I know that it is a popular game, but not everyone knows all the dynamics of the game (e.g. I didn’t remember the pipes move).
    b. The description of the RL agent used. Although the same approach is used in a previous paper, I usually prefer a self-standing paper (mostly for replication purposes).

**Best Paper Nomination:**

No

**Claims:**

The paper provides one main experiment to support the approach's ability to design better levels with RL agents as testers. The approach is interesting and a good fit for the workshop. However, I think the experimental setup lacks some important discussions.

1. My main concern is the “design” aspect of the approach. The only experiment the authors do is with a generic goal of “target score of 10”, but I am not sure this is enough to prove that LMMs can help design a level. Level design is very complex and qualitative, while the tested goal is very simple, made in a very simple game. It is unclear how the approach would scale with a more complex design goal. I would suggest that the authors do more experiments with more complex design goals, closer to actual real design settings (and for this, I would ask for support from actual game designers).
2. Because solving the cited design goal is actually pretty simple (the LMM could just create the first 10 pipes very distant to each other, and then close the gap), I would prefer having more qualitative analyses on the levels created by the approach and the baselines. From the results, it looks like all the baselines (except config-only) reach the target goal. However, it is unclear if the different approaches use different strategies to change the levels. Even with the same approach but different seed, the strategy for creating the level can change. I would suggest that the authors expand on this qualitative analysis.
3. It would be interesting to understand how the generated levels align with designers' desired target goals. My understanding is that this approach is targeted at game designers; however, these are not present in the evaluation. How well is the intention of a human designer represented by the generated levels (supposing that the target goal is defined by the human designer)?

**Suggestions:**

I already touched upon how the authors can improve the paper in the previous sections. In brief:

1. Expand the related work section and discussions with more PCG + RL and LMM works;
2. Expand the experiments section with more complex experiments, aligning with what human designers would do in a real setting; and
3. Test the approach in different games, more complex, where one can show the potential of automated design.

---

### Official Review · Reviewer_dXF3 · 2025-06-17
**Fly, Fail, Fix: A Novel Framework for AI-Driven Game Design Iteration**

**Rating:** 3
**Confidence:** 4

**Summary:**

This paper presents a novel framework that uses a large multimodal model (LMM) as a "game designer" and a reinforcement learning (RL) agent as a "playtester" to automatically iterate on and repair a game's design. The LMM adjusts game configuration parameters based on play traces from the RL agent to achieve a specified goal (e.g., a target player score). The authors test this framework on Flappy Bird, evaluating different feedback modalities (text summaries, gameplay images, or both) and demonstrating that the LMM can successfully refine broken game configurations to meet the design objective.

**Strengths:**

- The core idea of pairing an RL playtester with an LMM designer is innovative and presents a promising direction for AI-assisted game development.
- The experimental design is strong. The ablation study on feedback types (config-only, text, image, text+image) is well-executed and clearly demonstrates the necessity of gameplay feedback.
- The use of five distinct "broken" starting configurations to test the system's robustness is a commendable choice.
- The authors employ rigorous statistical analysis, such as using larger 50-episode rollouts for evaluation to overcome the high variance of the 5-episode feedback loop, and reporting Inter-quartile Mean (IQM) with confidence intervals.

**Weaknesses:**

- The choice of a fixed, pre-trained RL agent is a significant limitation. The authors note in their future work that the agent's performance degrades catastrophically with small physics changes. This brittleness questions whether the agent is a good proxy for a human player, or if the system is just learning to tune for the specific quirks of this one agent.
- The experimental task in Flappy Bird, aiming for a target score, may be too simple to demonstrate the full potential of the framework, particularly for visual feedback. Since horizontal progress is a direct proxy for score, it's unclear how the system would handle more nuanced gameplay goals where subtle visual cues might be more important than simple metrics.
- The paper contains a public GitHub link, which is a significant issue for a double-blind review process as it could compromise author anonymity. There are also minor typos, such as "abit" instead of "a bit" and "IWM" instead of "IQM" in a figure title.

**Best Paper Nomination:**

No

**Claims:**

The proposed "Fly, Fail, Fix" framework is thoughtfully designed and the experiments convincingly demonstrate its core viability. While the generality of the approach is somewhat limited by the brittleness of the chosen RL agent and the simplicity of the test-case game, the paper provides a strong foundation for future work

**Suggestions:**

same as above

---

### Official Review · Reviewer_YcsV · 2025-06-18
**Well-written paper with interesting ideas**

**Rating:** 4
**Confidence:** 4

**Summary:**

This paper focuses on game design and playtesting. The paper proposes to iteratively improve a game (according to a certain goal, for instance that the player achieves a score of 10) by having an artificial agent play the game and feeding the data of the interaction to an LMM that improves the game. The paper studies four methods (one baseline). In the baseline, the LMM does not receive interaction data, and in the other methods it receives text, image, or text and image. The finding is that all three methods, apart from the baseline, are able to achieve the game design goals and have indistinguishable performance.

It is a short paper (4 pages).

**Strengths:**

The paper is very clearly written and has enough details and experimental results. It provides a nice study on the use of LMMs as game designers integrated with automated player testing.

**Weaknesses:**

Perhaps that the reinforcement learning component is, as far as I can tell, limited (the player agent was already trained, and the LMM does not use any RL).

The conclusions could be made clearer earlier. There are also a couple of sentences that could be clearer in the Experiments section (3):
- what is the "broken configuration" (line 79);
- lines 88 to 92, about having run additional episodes after the experiments?

I only found a small typo: "trajecotries".

**Best Paper Nomination:**

No

**Claims:**

Yes, even though the claim could be a bit clearer and appear earlier (assuming the claim is that all three methods are able to achieve the goal and have indistinguishable performance).

**Suggestions:**

- Clarify the sentences that are unclear;
- Move the references, to appear before the supplementary material.

---

### Decision · Program_Chairs · 2025-06-19

**Decision:**

Accept

**Comment:**

This work introduces an automated game design framework that uses an RL agent for playtesting, together with a large multimodal model that edits the game based on the agent’s behavior. By analyzing play metrics and visual summaries, the model iteratively refines game mechanics to better align with design goals. The proposed approach was tested on Flappy Bird. The paper is well-written, and experimental results show the potential of the approach in real-world (game design) use-cases.
Reviewers recommend a few other highly relevant works to cite, point out typos, and suggest further clarification on the contributions, conclusions, and specific descriptions. We strongly encourage the authors to address these points in the camera-ready version.